# In Vivo Biofilm Formation on Novel PEEK, Titanium, and Zirconia Implant Abutment Materials

**DOI:** 10.3390/ijms24021779

**Published:** 2023-01-16

**Authors:** Andreas Wiessner, Torsten Wassmann, Johanna Maria Wiessner, Andrea Schubert, Bernhard Wiechens, Tristan Hampe, Ralf Bürgers

**Affiliations:** 1Department of Prosthodontics, University Medical Center Göttingen, 37075 Göttingen, Germany; 2Department of Orthodontics, University Medical Center Göttingen, 37075 Göttingen, Germany

**Keywords:** implant abutment, dental biomaterial, biofilm, in vivo study, zirconia, PEEK, titanium, peri-implantitis, biofilm management

## Abstract

The formation of biofilms on the surface of dental implants and abutment materials may lead to peri-implantitis and subsequent implant failure. Recently, innovative materials such as polyether-ether-ketone (PEEK) and its modifications have been used as abutment materials. However, there is limited knowledge on microbial adhesion to PEEK materials. The aim of this in vivo study was to investigate biofilm formation on the surface of conventional (titanium and zirconia) and PEEK implant abutment materials. Split specimens of titanium, zirconia, PEEK, and modified PEEK (PEEK-BioHPP) were manufactured, mounted in individual removable acrylic upper jaw splints, and worn by 20 healthy volunteers for 24 h. The surface roughness was determined using widefield confocal microscopy. Biofilm accumulation was investigated by fluorescence microscopy and quantified by imaging software. The surface roughness of the investigated materials was <0.2 µm and showed no significant differences between the materials. Zirconia showed the lowest biofilm formation, followed by titanium, PEEK, and PEEK-BioHPP. Differences were significant (*p* < 0.001) between the investigated materials, except for the polyether-ether-ketones. Generally, biofilm formation was significantly higher (*p* < 0.05) in the posterior region of the oral cavity than in the anterior region. The results of the present study show a material-dependent susceptibility to biofilm formation. The risk of developing peri-implantitis may be reduced by a specific choice of abutment material.

## 1. Introduction

The human oral cavity harbors a diverse and unique variety of more than 700 different microorganisms that normally organize themselves in complex structured biofilms [1,2]. These oral biofilms form immediately on any natural or artificial surface exposed to saliva and other oral fluids, which in turn serve as reservoirs for bacterial, viral, and fungal cells [3]. The eubiosis within biofilms may shift towards a predominance of disease-causing strains [4]. As a consequence, pathological biofilms attached to dental implants and implant-prosthetic abutments may, in the long term, lead to destructive inflammation of the peri-implant soft and hard tissues (i.e., peri-implant mucositis and peri-implantitis [5]), and in the worst case can result in the loss of the implant and the corresponding prosthetic superstructure [6,7,8]. Peri-implantitis, by definition associated with irreversible loss of surrounding bone, occurs frequently (overall implant based prevalence > 20%) [9,10] and is the main reason for serious complications in implant-retained prosthetic restorations [11,12,13,14,15].

Prosthetic materials used for implant abutments and removable or fixed reconstructions are of particular importance in the pathogenesis of peri-implant inflammation, because they are directly located at the biological weak point of the implant at the transition from peri-implant hard to soft tissue above the implant shoulder [16,17]. These parts of the implant superstructure are not enclosed by the alveolar bone like the implant itself, but directly exposed to the biological emergence profile, which additionally is the predilection site for the adhesion and accumulation of peri-implant biofilms [16,18,19,20,21,22]. Prosthetic abutments have a much more complex geometry than conventional tooth-supported prosthesis and are, therefore, even more difficult to access for oral hygiene by the patient, which in turn increases the potential of biofilm-associated infections [23,24,25]. Therefore, the development of novel anti-microbial and anti-adhesive implant abutment surfaces seems not only desirable, but essential [15,26,27,28,29].

In recent years, two biomechanically stable and biocompatible materials, in particular, have proven their worth as gold standard materials in implant-prosthodontics, namely titanium and zirconium oxide [30,31,32,33,34,35]. Both show reduced biofilm accumulation compared to other dental materials, mainly due to their bioinert properties and excellent polishability [17,31,36,37]. The urge of innovation by implant manufacturers and dedicated scientists to simultaneously improve osseointegration and reduce biofilm accumulation has led to an above-average number of promising novel implant-(prosthetic) materials and surfaces [38,39,40]. Polyether-ether-ketone (PEEK), a thermoplastic biocompatible polymer, has prevailed over other new developments in respect of mechanical/chemical resistance, biocompatibility, and low plaque affinity [41,42,43]. Therefore, PEEK has already been used commercially for some years in a wide variety of dental indications, and as an alternative to titanium and zirconia in clinical implantology [41,44,45,46]. Regarding biofilm accumulation, and apart from a small number of in vitro studies, surprisingly, no conclusive in vivo or clinical studies on microbial adhesion to PEEK are available [28,46,47,48].

Therefore, in the present study, we investigated the in vivo biofilm accumulation on four different implant abutment materials. In particular, the initial accumulation of microorganisms in the human oral cavity on two novel PEEK surfaces should be compared with that on titanium and zirconia, two well-proven implant-prosthetic materials (gold standards).

## 2. Results

### 2.1. Characterization of Test Surfaces

No statistically significant differences in surface roughness values Ra (*p* = 0.197) and Sa (*p* = 0.116) were found between any of the tested materials after high polishing (Table 1).

Widefield confocal micrographs also did not reveal any significant morphological differences between the four test substrata in the two- and three-dimensional display of both PEEK, the zirconia, or the titanium surfaces (Figure 1a,b).

### 2.2. In Vivo Biofilm Formation

The highest quantity of dental biofilms on the tested implant abutment surfaces was found for PEEK-BioHPP, where 19.7% (9.4%/25.3%; median and 25/75 percentiles) of the material surface was covered by oral biofilms, followed by PEEK-VestaKeep DC4430R with 17.81% (12.1%/24.1%), titanium with 11.1% (5.9%/15.7%), and zirconia with 6.5% (2.9%/9.6%) (Figure 2). Statistical analysis (linear mixed effect model) revealed significant differences in biofilm accumulation between titanium and the other three materials (*p* < 0.001). Subsequent post hoc analyses indicated statistically significant differences in microbial colonization between all tested materials (*p* < 0.001, respectively), except for the comparison between both PEEK materials (PEEK-BioHPP and PEEK-VestaKeep DC4430R, *p* = 0.953). Figure 3 shows representative fluorescent micrographs of the in vivo biofilm accumulations on the tested materials.

The multiple linear mixed effects model revealed significant differences in biofilm accumulation between the different intraoral localizations (canine vs. first molar) of the test specimens (*p* = 0.0123). Specimens located in the posterior region of the splints showed significantly higher overall biofilm adhesion (18.8%, median) than those positioned in the anterior region (8.9%). These differences were observed for all tested materials. Trend differences in overall biofilm accumulation were found between men and women, but they were not statistically significant (*p* = 0.079, Table 2).

## 3. Discussion

Microbial adhesion on implant-prosthetic substrata may cause peri-implant inflammation and therefore compromise long-term implant survival. This clinical trial investigated the in vivo plaque accumulation on four different implant abutment materials, with two established standard materials (titanium, zirconia) and two novel PEEK materials. To our knowledge, the present study is the first prospective clinical trial that compares initial in vivo biofilm accumulation between these implant abutment materials [30,31,32,33,34,35]. Previously conducted studies on biofilm accumulation on PEEK materials were performed in vitro throughout, and therefore significant clinical data are still missing. Although in vitro biofilm models may be used to generate initial indications of varying degrees of biofilm accumulation on different surfaces in a reproducible and manageable setting, it is not possible to simulate realistic conditions of a complex microflora or an interacting immune system of the host [49,50,51]. Therefore, in vitro results should always be verified by subsequent in vivo testing in the best case [52]. Due to the heterogeneous study setups and varying specimen preparation, existing in vitro studies on biofilm accumulation on PEEK, zirconia, and titanium led to deviating results [47,48]. Hahnel et al. showed significantly lower in vitro biofilm accumulation on PEEK than on titanium and zirconia abutments, whereas Barkarmo et al. did not find any significant differences in bacterial accumulation between PEEK and titanium [47,48].

Only for titanium, a surface roughness value of 0.2 µm (R_a_) has been established as a threshold below which no further influence of R_a_ on microbial accumulation is observed [53,54] In the present study, the surface roughness values (R_a_ and S_a_) of all tested material were below this threshold, with no statistically significant differences in all comparisons; therefore, the influence of roughness on biofilm formation in our clinical trial setup was eliminated. In biofilm testing, significant differences between the four implant-prosthetic materials were found, with the lowest biofilm accumulation on zirconia specimens (median covered surface: 6.5%), followed by titanium (median covered surface: 11.1%), and both PEEK materials (median covered surface PEEK-VestaKeep DC4430R: 17.51%; median covered surface PEEK-BioHPP: 19.7%). The differences between PEEK-VestaKeep DC4430R and PEEK-BioHPP were not significant. PEEK materials chemically belong to the polyaryletherketones (PAEK) and present favorable clinical properties such as high mechanical stability, biocompatibility, and chemical inertness [55,56]. Therefore, the application range of PEEK in restorative dentistry is growing rapidly and more and more clinical applications are being developed. In the present study, one conventional PEEK material (PEEK-VestaKeep DC4430R) and one PEEK material reinforced with 20% ceramic fillers (PEEK PEEK-BioHPP) were investigated, which represent the two most significant material specifications of PEEK. The addition of ceramic fillers is supposed to improve mechanical and aesthetic properties [57]. However, regarding biofilm accumulation, no significant difference between both materials was found. There are countless modifications of PAEK; therefore, the results of the present study cannot be applied to all materials, especially in comparison to titanium and zirconia, without any limitations. For example, in a recent in vivo study by Zeller et al., similar biofilm accumulation on a titanium-modified PAEK and zirconia were found [58]. In contrast, there are already some clinical investigations that compare microbial adhesion on titanium and zirconia. In most of these studies, no difference is found between titanium and zirconia, and if there is, zirconia seems to have a slightly lower potential for the accumulation of oral biofilms [31,59,60,61,62].

Additionally to the comparison between different implant-prosthetic substrata, we investigated the influence of the intra-oral localization of the specimens on biofilm formation. A significantly elevated biofilm accumulation in the posterior region was observed when compared to the canine position. These results agree with data from the recent literature [63,64]. The higher quantity of biofilm adhesion in the posterior regions of the oral cavity is associated with the localization of the excretory ducts of the large salivary glands, and a reduced manual cleaning by the soft tissue of the oral cavity [65,66].

Within the limitations of this study, we conclude that biofilm accumulation on implant-abutment substrata is material-dependent. The area covered by biofilm decreased in the following order: PEEK-BioHPP > PEEK-VestaKeep DC4430R > titanium > zirconia. All differences, except those found between both PEEK materials, were significant. In contrast to previous studies, this study is a prospective clinical trial using specimens with a standardized surface roughness below the threshold of 0.2 µm, and therefore the measured differences in biofilm formation are most likely influenced by the material and not by general surface characteristics. The risk of developing peri-implantitis may be reduced by a specific choice of abutment material. Future research could focus on biomaterials as such, or the direct alteration of the oral microbiome. The former might be achieved by the development of abutment materials with active antimicrobial effects that can inhibit bacterial growth, the latter by the domiciliary use of pro- or postbiotics and ozonized water to alter or to eradicate biofilms [67,68,69].

## 4. Materials and Methods

### 4.1. Preparation and Characterization of the Test Specimens

In order to achieve a high level of comparability between the four implant abutment materials in biofilm testing, special split specimens were developed. Firstly, square rods (height/length/width = 30.0/2.0/2.0 mm^3^) were produced from the test materials (titanium, zirconia, PEEK-BioHPP, and PEEK-VestaKeep DC4430R, see Table 3) strictly according to the manufacturer’s instructions. These rods were bonded together on their long sides with a dental luting composite (Panavia 21, Kuraray Noritake Dental Inc., Okayama, Japan) and sawed into specimens of equal height (height/length/width = 1.0/4.0/4.0 mm^3^, see Figure 4) with a diamond saw (Exakt 300, Exakt GmbH, Norderstedt, Germany).

Specimens (n = 80) were polished to a high gloss in order to reduce the influence of surface morphologies on biofilm formation by using a standardized polishing process with silicon carbide grinding paper with descending abrasiveness (500, 800, 1200, and 4000) and an automated polishing machine (Exakt 400 CS, Exakt GmbH, Norderstedt, Germany). The arithmetic mean roughness (R_a_) and the area-related mean arithmetic height (S_a_) were calculated via widefield confocal microscopy (Zeiss Smartproof 5, Carl Zeiss, Jena, Germany) and automated software analysis (ConfoMap ST 7.4.8076, Carl Zeiss, Jena, Germany).

Specimens were disinfected by ultrasonication in 3% sodium hypochlorite for 20 min and then washed in distilled water before further processing. For each study object, four split specimens were fixed to an individual removable acrylic upper jaw splint (Figure 5) used to position the specimens in the buccal region of the canines and first molars (teeth 13, 16, 23, and 26, respectively).

### 4.2. In Vivo Biofilm Formation

The present study was conducted as a prospective clinical trial. The subject collective included ten women and ten men (age 22 to 34 years, mean 25.7 years, all healthy and non-smokers). The cohort size was chosen according to a prior in vivo study of our research group [70]. None of the volunteers had used antibacterial mouth rinses or systemic antibiotics in the two months prior to the start of the study. All participants had excellent oral hygiene and no periodontal diseases (plaque indices < 15%, sulcus bleeding indices < 10%) and no caries lesions. The oral examination was carried out by an experienced dentist. Informed written consent had been given by all subjects, and the study had been approved by the Ethics Committee of the Faculty of Medicine, University Medical Center Göttingen (application number 17-7-15).

The subjects were instructed to insert and remove their splint only for oral hygiene measures and food or beverage intake. The splints were worn for 24 h. Then, the plaque-covered specimens were carefully detached under sterile conditions and immediately processed for fluorescence staining.

### 4.3. Visualization and Quantification of Adhering Biofilms

Specimens were transferred to 24-well-plates, fixed in the wells by duplicating silicone (Z-Dupe Shore A-20, Henry Schein Inc., Melville, NY, USA) and washed threefold in PBS to remove non-adhered cells. The fluorescence dye Hoechst 33342 (Sigma-Aldrich, St. Louis, MO, USA) was used to quantify adhering biofilms (fluorescence emission maximum for approximately 461 nm; excitation maximum of 355 nm). The fluorescence staining solution was prepared by diluting 5 µL of the fluorescence stain (1 mg/mL) in one milliliter of sterile 0.85% sodium chloride (Merck KGaA, Darmstadt, Germany) for each well. The specimens were incubated in the staining solution for 13 min. After washing with 1 mL of sterile 0.85% sodium chloride, the stained biofilms were fixed with 2% paraformaldehyde (PFA) (Sigma Aldrich, St. Louis, MO, USA). Each specimen was carefully positioned on a coverslip and stored in the dark at 4 °C until further processing.

Fluorescence emission was determined with a fluorescence microscope (Keyence BZ-X710, Keyence Corporation, Osaka, Japan) in combination with an image processing software (BZ-X Analyzer, Keyence Corporation, Osaka, Japan). In each specimen, the fluorescent microscopic images of three randomly selected sites on each of the four test substrata were captured (20× magnification). Thus, 12 images were obtained for each specimen and with a number of 20 subjects, a total of 960 images were obtained. The areas covered by cells were calculated as the percentage of specific standard microscopic fields (500 µm × 750 µm = 0.375 mm^2^) with the image analysis software Image J 1.51k Fiji (National Institute of Health, MD, USA).

### 4.4. Statistical Analysis

Calculations were performed with statistical software R (version 3.4.0, The R Foundation of Statistical Computing, Vienna, Austria). The significance level was set to α = 0.05%. A Box–Cox transformation was applied to the dependent variable ‘area in percent’. Titanium was set to be the reference material. A linear mixed effect model was used to assess the influence of material, intraoral position, as well as participants’ gender and age on the quantitative plaque accumulation. To determine topographical differences in the specimens’ surfaces, data from the roughness measurements were analyzed using a two-way ANOVA. Subsequently significant differences were calculated using post hoc analyses by Tukey.

## Figures and Tables

**Figure 1 ijms-24-01779-f001:**
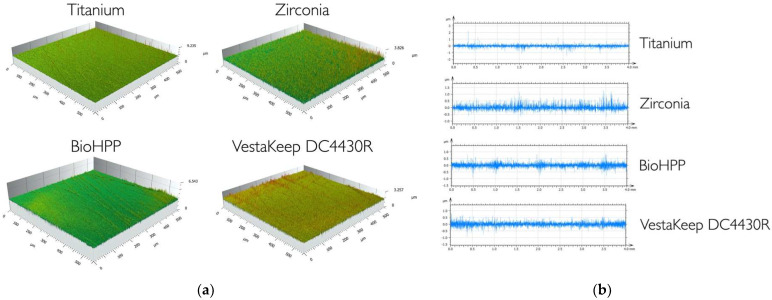
(**a**). Widefield confocal micrographs of the test materials, 3D-profile (500 × 500 µm^2^). (**b**). Widefield confocal micrographs of the test materials, 2D-profile (40 mm).

**Figure 2 ijms-24-01779-f002:**
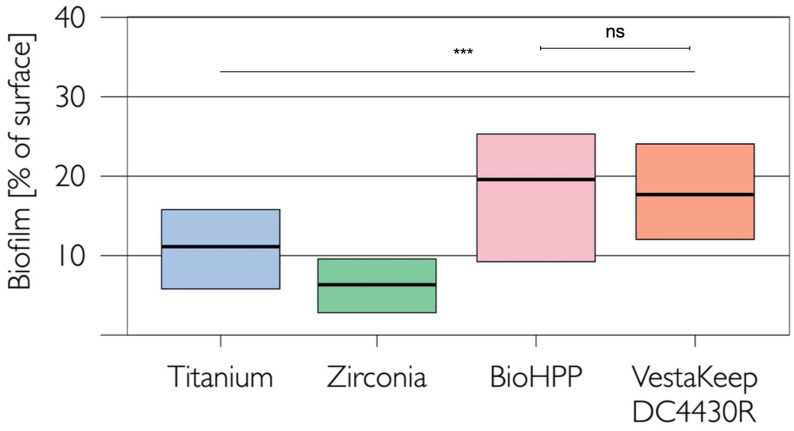
Coverage [% of the of specific standard microscopic fields (500 µm × 750 µm] with in vivo biofilm after 24 h on four test implant abutment materials (medians and 25/75 percentiles), ns = not significant, *** = *p* < 0.001.

**Figure 3 ijms-24-01779-f003:**
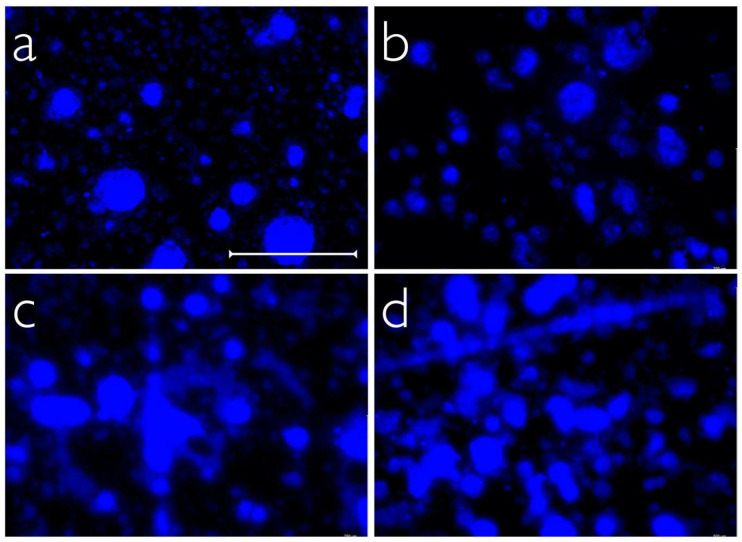
Exemplary fluorescence micrographs of titanium (**a**), zirconia (**b**), PEEK-BioHPP (**c**), and PEEK-VestaKeep DC4430R (**d**) after 24 h of in vivo biofilm formation and fluorescence staining (Hoechst 33342), scale bar equals 250 µm.

**Figure 4 ijms-24-01779-f004:**
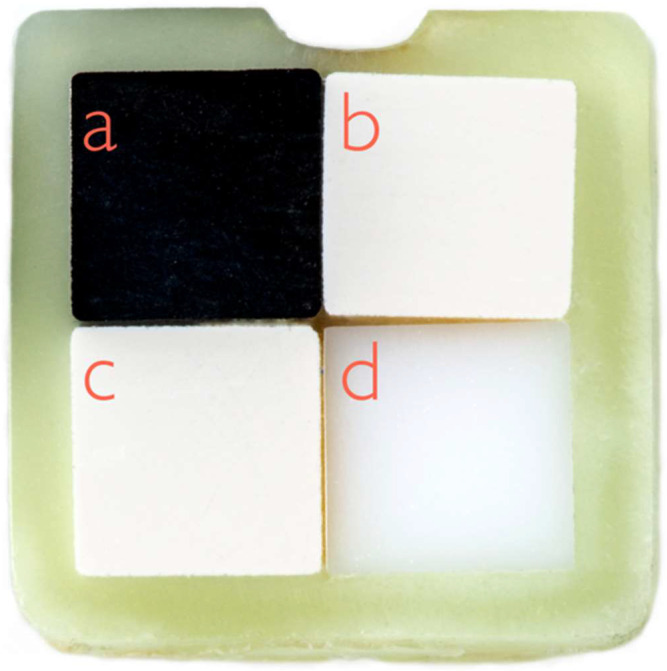
Split specimens consisting of four implant abutment test substrata (each 4 mm × 4 mm): titanium (**a**), zirconia (**b**), PEEK-BioHPP (**c**), and PEEK-VestaKeep DC4430R (**d**).

**Figure 5 ijms-24-01779-f005:**
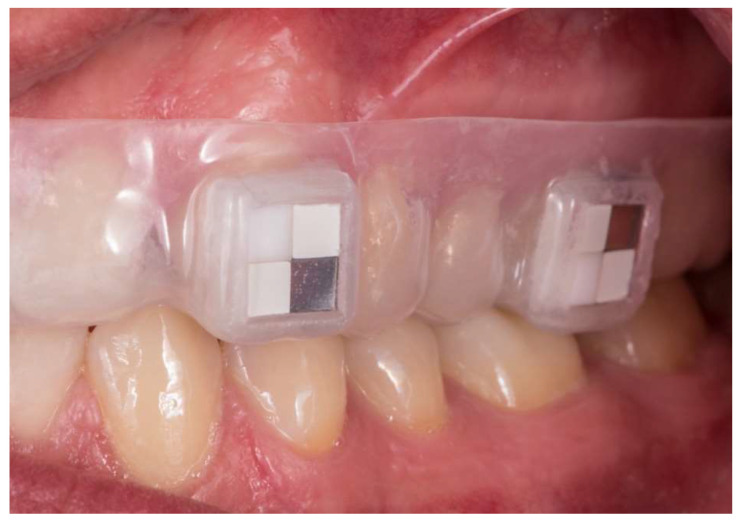
Individual splint in situ with two split specimens on the canine and first molar of the left side of the upper jaw.

**Table 1 ijms-24-01779-t001:** The arithmetic mean roughness (R_a_) and the area-related mean arithmetic height (S_a_) of the test materials (medians and 25/75 percentiles); no significant differences were found.

	Surface Roughness [μm]	
Material	R_a_	S_a_
PEEK-BioHPP	0.099 (0.086/0.114)	0.133 (0.114/0.152)
PEEK-VestaKeep DC4430R	0.100 (0.085/0.121)	0.130 (0.108/0.168)
Titanium	0.114 (0.087/0.128)	0.130 (0.111/0.150)
Zirconia	0.100 (0.090/0.112)	0.117 (0.103/0.130)

**Table 2 ijms-24-01779-t002:** Biofilm formation [% of the area] on four different test materials (medians and 25/75 percentiles) in correlation with the localization of the test specimens and the gender of the subjects. Significant differences (*p* < 0.05) are indicated by equal letters.

	Intraoral Position		Gender	
	Canine	First Molar	Female	Male
Titanium	5.3 (1.5/12.8) ^a^	11.6 (6.3/24.3) ^a^	15.1 (10.6/20.6)	7.7 (5.4/13.2)
Zirconia	1.9 (1.1/5.6) ^b^	5.4 (3.5/18.0) ^b^	8.6 (3.2/15.1)	3.6 (3.0/8.4)
PEEK-BioHPP	14.2 (3.8/19.4) ^c^	19.6 (9.0/44.9) ^c^	21.5 (14.9/34.9)	17.7 (9.3/23.4)
PEEK-VestaKeep DC4430R	8.7 (4.2/19.3) ^d^	22.5 (10.7/38.8) ^d^	21.2 (17.0/25.3)	15.5 (11.4/21.5)

**Table 3 ijms-24-01779-t003:** Test implant abutment materials used in this study.

Class of Material	No.	Test Material	Manufacturer
Titanium (grade 2)	1	Zenotec Ti pur	Wieland Dental + Technik GmbH & Co. KG, Pforzheim, Germany
Zirconia (zirconium dioxide)	2	Cercon base	Dentsply Sirona, Charlotte, NC, USA
PEEK (polyetheretherketone)	3	BioHPP	Bredent GmbH & Co. KG, Senden, Germany
	4	VestaKeep DC4430R	Evonik Industries AG, Essen, Germany

## Data Availability

Not applicable.

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
