# Peer review of "In Vivo Biofilm Formation on Novel PEEK, Titanium, and Zirconia Implant Abutment Materials"

_ijms, 2023, doi:10.3390/ijms24021779_

Round 1

Reviewer 1 Report

Implants should improve osseointegration and reduce biofilm accumulation or kill bateira in the adhered biofilms.

The authors should present the antibatrial properties of the materials tested (if any). The existence of adhered biofilms do not allow to ascertain whether cells in the biofilms are alive or dead.

There is extensive literature on oxide photocatalytic effects (eg TiO; ZnO; MgO, etc). Hydroxyl anion and water can be oxidized to OH free radicals with strong oxidation capacity. These activated surfaces can inhibit and kill microorganisms.

Please, answer the following important question. In the biofilms, are the microorganisms alive or dead?

Search for antimicrobial effects fo each material or perform additional experiments.

Author Response

Thank you for your helpful remarks!

"The authors should present the antibatrial properties of the materials tested (if any). The existence of adhered biofilms do not allow to ascertain whether cells in the biofilms are alive or dead."

Since the materials tested are commercially available, we did not test them for antibacterial properties - which are not present to our knowledge. We tested those "passive", clinically already widely used materials for the direct detection of different biofilm accumulation in order to be able to issue an immediate clinical recommendation for action. 

"There is extensive literature on oxide photocatalytic effects (eg TiO; ZnO; MgO, etc). Hydroxyl anion and water can be oxidized to OH free radicals with strong oxidation capacity. These activated surfaces can inhibit and kill microorganisms."

The effects you describe are indeed the subject of current research and a very promising approach for further studies of our research group. However, the aim of the present study was not to develop novel surfaces or to activate antimicrobial properties. Instead, we tested commercially available materials, that are relevant for the dental practitioner. Our study is the first one to asses four materials simultaneously under in vivo conditions and - due to the special specimen design - without interfering influences of different intraoral positioning. We added a sentence in the discussion section stating that future research is required for the development of activated surfaces that inhibit microorganisms. The sentence reads as follows: "Future research could additionally focus on the development of abutment materials with active antimicrobial effects that can inhibit bacterial growth."

"Please, answer the following important question. In the biofilms, are the microorganisms alive or dead?

Search for antimicrobial effects fo each material or perform additional experiments."

In addition to purely physicochemical, passive surface properties, active, antimicrobial surfaces can influence, reduce or prevent bacterial growth - the present study is focused on the first. The physicochemical properties of the four materials, even without surface activation, lead to significant differences in biofilm accumulation - as we could show.
Bacterial growth in the biofilm was therefore measured as a whole - adherence of bacteria requires primarily living microorganisms, an antimicrobial effect that inactivates or kills bacteria in an already formed biofilm was not suspected and consequently not investigated.

Please see the attachment, changes are marked red.

Reviewer 2 Report

Manuscript of considerable interest for the dental sector, before the evaluation of a possible publication.

Abstract, to better highlight the results obtained.

Keywords: few add specific ones

Introduction add reference to the new classification of peri-implant disease, highlighting how to maintain a constant state of eubiosis even in intact implants as already studied by the research group of Prof scribante.

10.3390/microorganisms10040675 10.3390/app12073250

Results, very confusing, highlight the statistically significant data

Materials and methods well described, how the sample size was calculated.

Discussion: add home maintenance through the use of postbiotics and ozonized water as future objectives.

10.3390/app12062800

Conclusions: add proactive action

Author Response

"Manuscript of considerable interest for the dental sector, before the evaluation of a possible publication."

Thanks for your kind and helpful review!

"Abstract, to better highlight the results obtained."

In the abstract, we rephrased the result section and added information on the levels of significance. The section now reads: "Differences were significant (p < 0.001) between the investigated materials, except for the polyether-ether-ketones. Generally, biofilm formation was significantly higher (p < 0.05) in the posterior region of the oral cavity than in the anterior region."

"Keywords: few add specific ones"

We added more keywords: "implant abutment; dental biomaterial; biofilm; in vivo study, zirconia, PEEK, titanium, peri-implantitis, biofilm management."

"Introduction add reference to the new classification of peri-implant disease, highlighting how to maintain a constant state of eubiosis even in intact implants as already studied by the research group of Prof scribante."

We added the suggested references to the introduction. Also, we adjusted the introduction accordingly: "The eubiosis within biofilms may shift towards a predominance of disease-causing strains [4]."

Results, very confusing, highlight the statistically significant data

As suggested, we added additional explanations in the figure legends and modified figure 2 and table 2. 

Materials and methods well described, how the sample size was calculated.

We used the sample size of a prior study, which is now also cited. We considered the group size to be a good compromise between manageability and sufficient number. Moreover, the production of the special specimens was very laborious and therefore, with four specimens per study participant, practically limited the group size. 

"Discussion: add home maintenance through the use of postbiotics and ozonized water as future objectives.

Conclusions: add proactive action"

Thank you for your helpful input. We added the suggested information and are confident that they improved the quality of the manuscript: "Future research could focus on biomaterials as such or the direct alteration of the oral microbiome. The former might be achieved by the development of abutment materials with active antimicrobial effects that can inhibit bacterial growth, the latter by the domiciliary use of pro- or postbiotics and ozonized water to alter or to eradicate biofilms [67–69]."

Please see the attachment, changes are marked red.

Reviewer 3 Report

The study design is good in terms of equal treatment and equal conditions for individual materials.

Only, in my opinion, the expression time is very short. I recommend a similar study, but with a longer period of time for samples in the oral cavity, which would lead to greater reliability of the study.

Round 2

Reviewer 1 Report

I do not think this manuscript is inside the scope of IJMS.

The relationship between structure and function of the materials is not the focus of this technological research that should be evaluated by a more especialized Dentistry journal. The data do not provide any new concept or insight.

Reviewer 2 Report

the manuscript was properly revised based on the comments that were suggested in the first session